# Influence of Inherited Seizure Susceptibility on Intermittent Voluntary Alcohol Consumption and Alcohol Withdrawal Seizures in Genetically Epilepsy-Prone Rats (GEPR-3s)

**DOI:** 10.3390/brainsci14020188

**Published:** 2024-02-19

**Authors:** Gleice Kelli Silva-Cardoso, Prosper N’Gouemo

**Affiliations:** Department of Physiology and Biophysics, Howard University College of Medicine, Washington, DC 20059, USA; gleicekelli.cardoso@howard.edu

**Keywords:** genetic epilepsy, seizures, sex, alcohol intake, alcohol preference, rat

## Abstract

Background: The link between epilepsy and alcohol consumption is complex, with conflicting reports. To enhance our understanding of this link, we conducted a study to determine how inherited seizure susceptibility affects voluntary alcohol consumption and influences alcohol withdrawal seizures in male and female genetically epilepsy-prone rats (GEPR-3s) compared to Sprague Dawley (SD) rats. Methods: In the first experiment, animals were given access to two bottles simultaneously, one containing water and the other 7.5%, 15%, or 30% (*v*/*v*) alcohol three times a week for each dose after acclimation to drinking water. In a second experiment, animals were tested for acoustically evoked alcohol seizures 24 h after the last session of voluntary alcohol consumption. Results: Analysis revealed that GEPR-3s (males and females) had lower alcohol intake and preference than SD rats, particularly at lower alcohol concentrations. However, female GEPR-3s consumed more alcohol and had a higher alcohol preference than males. Furthermore, withdrawal from voluntary alcohol consumption facilitated the onset and duration of seizures in GEPR-3s. Conclusions: Our study suggests that genetic seizure susceptibility in GEPR-3s is negatively associated with alcohol consumption. However, withdrawal from low to moderate amounts of alcohol intake can promote epileptogenesis in the epileptic GEPR-3s.

## 1. Introduction

Epilepsy is one of the most common neurological disorders characterized by recurrent seizures, and alcohol is a commonly abused substance. Evidence indicated that alcohol use can increase the risk of developing seizures, especially during withdrawal from chronic alcohol intake [1,2]. Additionally, patients with a genetic susceptibility to generalized epilepsy may be at increased risk for seizures related to excessive alcohol consumption [3]. However, the relationship between alcohol consumption and seizures is complex, with conflicting reports. Studies reported that patients with epilepsy might drink more than the general population and have a higher risk of developing seizures associated with alcohol use [4,5,6,7]. In contrast, consumption of small to moderate amounts of alcohol does not seem to increase seizure frequency in non-alcoholic epileptic patients [8,9]. A preclinical study reported that alcohol consumption and subsequent withdrawal transiently increased the frequency of spontaneous recurrent seizures in the pilocarpine post-status epilepticus model of temporal lobe epilepsy, suggesting a positive link between limbic seizures and alcohol intake [10]. Furthermore, alcohol withdrawal seizure-resistant mice consumed more alcohol than withdrawal seizure-prone mice [11]. Additionally, mice from the withdrawal seizure-prone F1 selected line consumed more alcohol than alcohol withdrawal seizure-resistant F1 mice, irrespective of sex [12]. Interestingly, mice from the alcohol withdrawal seizure-resistant F2 selected line consumed more alcohol than withdrawal seizure-prone F2 mice, but this effect was only observed in females [12]. These findings suggest that the link between alcohol withdrawal seizure susceptibility and alcohol consumption is multifaced with limited shared genetics. Although inbred mice have been used in evaluating genetic influences on alcohol consumption, rat strain-selected lines can model multiple facets of addiction and alcohol consumption [13,14].

The genetically epilepsy-prone rat (GEPR) is a valid model for studying the mechanisms underlying neuronal hyperexcitability that enhances inherited seizure susceptibility [15,16]. The GEPR was derived from Sprague Dawley (SD) rats, and two substrains of the GEPR have been characterized based on the phenotype of acoustically evoked seizures [17]. The most severe seizure substrain is GEPR-9, which exhibits a tonic extension of both forelimbs and hindlimbs (complete tonic seizures), and the moderate seizure substrain is GEPR-3, which exhibits generalized tonic–clonic seizures (GTCSs) followed by tonic extension of the forelimb (or partial tonic seizures) in severe cases [17]. Interestingly, female GEPR-9s and GEPR-3s exhibit greater seizure susceptibility than their male counterparts [16,17,18]. GEPR-3s were explicitly included in this study because they exhibit acoustically evoked GTCSs that are similar to the most common type of acoustically evoked seizures following withdrawal from severe alcohol intoxication [18,19]. However, the extent to which inherited seizure susceptibility in GEPR-3s alters alcohol consumption and withdrawal seizures remains unknown. Nevertheless, a non-epileptic strain derived from SD rats had lower alcohol intake; such an effect may also occur in GEPR-3s [20,21].

Our study aims to investigate the influence of the genetic seizure trait on short-term voluntary alcohol consumption and alcohol withdrawal-induced seizures in male and female GEPR-3s compared to non-seizure-prone SD rats.

## 2. Materials and Methods

### 2.1. Animals

We used a total of 96 rats for the study. We used 8-week-old GEPR-3s (12 males and 12 females) and SD rats (12 males and 12 females) for the voluntary ethanol consumption experiments. Additionally, 28-day-old GEPR-3s (12 males and 12 females) and SD rats (12 males and 12 females) were used to determine whether the litters had seizure susceptibility. The GEPR-3s and SD rats were obtained from our animal colony at Howard University College of Medicine, Washington D.C., USA. The GEPR-3s used in this study belong to the 45th generation of our colony. We ensure the presence of the genetic seizure trait in our colony by breeding only one male and one female GEPR-3 that have exhibited acoustically evoked GTCSs. Post-weaning, on their 21st day, GEPR-3s and SD rats were housed in same-sex pairs per cage in a room with controlled temperature and humidity and subjected to a 12 h/12 h light/dark cycle, with unlimited access to food and water. All GEPR-3s in our colony exhibited acoustically evoked seizure susceptibility. The GEPR-3s and SD rats subjected to voluntary ethanol drinking were selected randomly from six litters, with two males and two females per litter. Adult GEPR-3s and SD rats were housed individually for one week (acclimation) before starting the ethanol-drinking experiments. We made all possible efforts to minimize the number of animals used in experiments and any discomfort experienced by the animals. All experimental procedures were approved by the Institutional Animal Care and Use Committee (Protocol MED 20-04), following the guidelines set by the National Institutes of Health Guide for the Care and Use of Laboratory Animals [22].

### 2.2. Voluntary Ethanol Consumption Paradigm: Intermittent Alcohol Access Two-Bottle Choice Paradigm

Ethanol (7.5%, 15%, and 30%, *v*/*v*) solutions were prepared in purified water from ethanol 95% stock solution (U.S.P., The Warner-Gram Company, Cockeysville, MD, USA). All fluids were presented in 40 mL graduated Drinko-measurer bottles (Amuza, San Diego, CA, USA) with stainless steel sippers inserted through two eyelets at the front of the cage. Bottles were weighed before and 24 h after fluid presentation, and the body weight of each rat was measured daily. We used the voluntary intermittent alcohol access two-bottle choice paradigm, which allowed the animals to have access to ethanol without sweeteners during three sessions of 24 h per week [23,24]. After one week of acclimatization in their home cages, GEPR-3s and SD rats were offered water from two Drinko-measurer bottles for two weeks to avoid preference or aversion behavior for a specific bottle [11]. All animals were given 24 h access to two bottles, one with 7.5%, 15%, or 30% (*v*/*v*) ethanol and the other with water, starting Monday at 10.00 a.m. After 24 h, the ethanol bottle was replaced with another bottle of water that was available for the next 24 h. This pattern was repeated on Wednesdays and Fridays. However, the rats were given unlimited access to two water bottles on Tuesdays, Thursdays, Saturdays, and Sundays. To ensure unbiased results, the placement of ethanol bottles was alternated on Wednesdays and Fridays to control for lateral preferences. After consuming 7.5% or 15% ethanol, rats were given access to water only for seven days the following week (as shown in Figure 1). This washout period was implemented to minimize any interference when measuring the animals’ ethanol consumption towards a different ethanol concentration.

### 2.3. Acoustically Evoked Seizure Testing

The GEPR-3s and SD rats selected for voluntary ethanol drinking and seizure testing were not exposed to any acoustic stimulus between postnatal day 21 (P21) and 28 (P28). The remaining GEPR-3s and SD rats (two males and two females per litter) were subjected to three acoustically evoked seizure tests between P21 and P28 to determine the litter’s seizure susceptibility. Twenty-four hours following the last ethanol or water consumption session, GEPR-3s and SD rats were placed in an acoustic chamber (Med Associates, St. Albans, VT, USA) and tested for acoustically evoked seizure susceptibility (Figure 1). The sound stimulus (100–105 decibels of pure tones) was applied for 60 s or until seizure onset. The phenotype of seizures was classified as follows [18]: stage 0, no seizures in response to an acoustic stimulus; stage 1, wild running seizures (WRSs); stage 2, two or more episodes of WRSs; stage 3, one episode of WRS followed by GTCS characterized by tonic dorsiflexion of the neck, tonic flexion of shoulder, and bouncing clonic seizures (or clonus, i.e., tonic–clonic seizures while the animal is lying on its belly); stage 4, two episodes of WRSs followed by GTCSs; and stage 5, one episode of WRS followed by GTCS and tonic forelimb extension (FLE, partial tonic seizures).

### 2.4. Data Analysis

The investigators were blinded to group allocation during experiments and data analysis, and Origin 2023b 10.05.157 software (OriginLab Corporation, Northampton, MA, USA) was used for statistical analyses and to create graphs. For ethanol consumption, dependent measures included ethanol dose (g/kg), alcohol preference (%), and water intake (mL/kg). The body weight of each rat was used to calculate grams of ethanol intake per kilogram of body weight and milliliter of water intake per kilogram. Preference for ethanol over water (the ratio of ethanol to total fluid intake) was calculated at the 24 h time point. The measurements were realized on Monday, Wednesday, and Friday and were used as averages for calculating alcohol intake, ethanol preference, and water intake. For seizure testing, the incidences of WRSs, GTCSs, and FLE were recorded for each group. The time interval from the start of the acoustic stimulus to the onset of the first episode of WRSs was recorded as the seizure latency (or seizure onset). The Fisher Exact test was used to analyze the WRSs, GTCSs, and FLE incidences. The seizure severity was analyzed using the Kruskal–Wallis test. Before ANOVA was performed, data were subjected to the Kolmogorov–Smirnov test for normality and Levene’s test for homogeneity of variances. The ethanol intake (g/kg), preference (%), and water intake (mL/kg) were first analyzed using three-way ANOVA with strain and sex as a within-subject factor and ethanol concentrations (7.5%, 15%, and 30%) as a between-subject factor. For each ethanol concentration, data were then analyzed using two-way ANOVA with strain as a between-subject factor and sex as a within-subject factor. The seizure latency and duration were analyzed using one-way ANOVA followed by the Tukey post hoc correction that compares the difference between each pair of means of the groups with an appropriate adjustment for the multiple testing (q); the cut-off for statistical significance was set at *p* < 0.05. Data are presented as percentages (%) for ethanol preference and seizure incidences (WRSs, GTCSs, FLE); mean ± S.E.M. for ethanol intake, water intake, seizure latency, and seizure duration; or median ± median average deviation for seizure severity.

## 3. Results

### 3.1. Effects of Strain, Sex, and Ethanol Concentrations on Voluntary Ethanol Consumption

We first analyzed the impact of strain, sex, and alcohol concentrations (7.5%, 15%, and 30%) and their interactions. Three-way ANOVA revealed significant mean differences for strain (F_1,132_ = 19.96, *p* < 0.0001), sex (F_1,132_ = 45.92, *p* < 0.0001), and ethanol concentrations (F_2,132_ = 20.70, *p* < 0.0001). Additionally, there was a significant interaction between strain and ethanol concentrations (F_2,132_ = 3.291, *p* = 0.04) but not between strain and sex (F_1,132_ = 0.122, *p* = 0.726) or sex and ethanol concentrations (F_2,132_ = 2.328, *p* = 0.101), or strain, sex, and ethanol concentrations (F_2,132_ = 0.101, *p* = 0.903). Furthermore, there were significant interactions between male GEPR-3s and male SD rats (q = 5.03, *p* = 0.003) and between female GEPR-3s and female SD rats (q = 4.30, *p* = 0.015). In addition, there were significant interactions between male GEPR-3s and female GEPR-3s (q = 7.44, *p* < 0.001) and between male SD rats and female SD rats (q = 6.71, *p* < 0.0001). A significant interaction was also observed between GEPR-3s and SD rats during the 7.5% alcohol test (q = 6.85, *p* < 0.0001), but not for higher ethanol concentrations.

### 3.2. Effects of Voluntary Ethanol Consumption: 7.5% Alcohol Concentration

Next, we analyzed the effects of strain and sex on ethanol intake, preference, and water intake. For ethanol intake, two-way ANOVA revealed significant differences in the strain (F_1,44_ = 11.15, *p* = 0.0017) and sex (F_1,44_ = 16.08, *p* = 0.0002); no statistical difference was found in the interaction between strain and sex (F_1,44_ = 0.004, *p* = 0.94). Interestingly, female GEPR-3s consumed significantly more 7.5% ethanol than male GEPR-3s (q = 3.94, *p* = 0.03, Figure 2A). Similarly, female SD rats consumed significantly more 7.5% ethanol than males (q = 4.08, *p* = 0.02, Figure 2A) and male GEPR-3s (q = 7.35, *p* = 0.0001, Figure 2A). No statistical differences were observed between male and female GEPR-3s compared to male and female SD rats (Figure 2A). For ethanol preference, two-way ANOVA revealed a significant difference in the strain (F_1,44_ = 10.25, *p* = 0.002) but not in sex (F_1,44_ = 2.37, *p* = 0.1307) or the interaction between strain and sex (F_1,44_ = 0.99, *p* = 0.32). Male GEPR-3s had a significantly lower preference for 7.5% ethanol than male SD rats (q = 4.19, *p* = 0.02, Figure 3B); no statistical difference was found between female GEPR-3s and female SD rats (Figure 3B). Furthermore, no statistical difference was observed in ethanol preference between female and male GEPR-3s (q = 2.53, *p* = 0.29, Figure 2B) or female and male SD rats (q = 0.54, *p* = 0.98). We also analyzed water intake in GEPR-3s and SD rats. Two-way ANOVA revealed no statistical differences in the strain factor (F_1,44_ = 2.51, *p* = 0.11), sex factor (F_1,44_ = 2.30, *p* = 0.135), and interaction between strain and sex (F_1,44_ = 3.50, *p* = 0.07). Female GEPR-3s did not exhibit a statistical difference in water intake compared to male GEPR-3s in the 7.5% ethanol test (q = 3.39, *p* = 0.09, Figure 2C). Similarly, the analysis did not show statistical differences between male SD male and female SD rats (q = 2.65, *p* = 0.25). No statistical differences were observed for male or female GEPR-3s compared to male or female SD rats (Figure 2C).

### 3.3. Effects of Voluntary Ethanol Consumption: 15% Alcohol Concentration

We further evaluated the effects of strain and sex on ethanol intake, preference, and water intake during the 15% ethanol test. Two-way ANOVA revealed significant differences in the strain factor (F_1,44_ = 5.01, *p* = 0.03) and sex factor (F_1,44_ = 18.11, *p* = 0.0001), but not in the interaction between strain and sex (F_1,44_ = 0.03, *p* = 0.84). Quantification of ethanol intake revealed that female GEPR-3s consumed significantly more 15% ethanol than male GEPR-3s (q = 4.45, *p* = 0.01, Figure 2D). No statistical differences were found between male and female GEPR-3s compared to male and female SD rats (Figure 2D). For ethanol preference, the analysis showed significant differences in the strain factor (F_1,44_ = 6.37, *p* = 0.015) and sex factor (F_1,44_ = 6.96, *p* = 0.011), but not in the interaction between strain and sex (F_1,44_ = 1.60, *p* = 0.21) in the 15% ethanol test. Further analysis revealed that female GEPR-3s had a higher 15% ethanol preference than male GEPR-3s (q = 3.90, *p* = 0.04, Figure 2E), while no change was found between male and female SD rats (q = 1.37, *p* = 0.76, Figure 2E). Moreover, male GEPR-3s have a lower preference for the 15% alcohol than male SD rats (q = 3.79, *p* = 0.04, Figure 2E). No statistical differences were observed between the females of the two strains. We also evaluated water intake during the 15% ethanol test. Two-way ANOVA revealed significant differences in the sex factor (F_1,44_ = 15.96, *p* = 0.0002) but not in the strain factor (F_1,44_ = 0.01, *p* = 0.89) or the interaction between strain and sex (F_1,44_ = 1.79, *p* = 0.18) in the 15% ethanol test. Further analysis revealed that male GEPR-3s had a significantly higher water intake than female GEPR-3s (q = 5.33, *p* = 0.002, Figure 2F), while no statistical differences were observed between male and female SD rats (q = 2.65, *p* = 0.25, Figure 2F) or for the male or female GEPR-3s compared to male or female SD rats.

### 3.4. Effects of Voluntary Ethanol Consumption: 30% Alcohol Concentration

We also evaluated the effects of strain and sex on ethanol intake, preference, and water intake during the 30% ethanol test. Two-way ANOVA revealed significant differences in the strain factor (F_1,44_ = 4.09, *p* = 0.04) and sex factor (F1,44 = 18.53, *p* < 0.001) but not in the interaction between strain and sex (F_1,44_ = 0.81, *p* = 0.37). Subsequent comparisons revealed that female GEPR-3s consumed significantly more 30% ethanol than male GEPR-3s (q = 3.68, *p* = 0.003, Figure 2G), while no statistical difference was observed in female SD rats compared to males (q = 3.40, *p* = 0.09, Figure 2G). No statistical differences were found in male or female GEPR-3s compared to male or female SD rats (Figure 2G). For ethanol preference during the 30% ethanol test, two-way ANOVA revealed significant differences in the strain factor (F_1,44_ = 9.97, *p* = 0.002), sex factor (F_1,44_ = 5.55, *p* = 0.022), and interaction between strain and sex (F_1,44_ = 4.38, *p* = 0.04). Further comparisons showed that female GEPR-3s had a significant preference for 30% ethanol compared to male GEPR-3s (q = 4.44, *p* = 0.015, Figure 2H), while no change was observed between male and female SD rats (q = 0.26, *p* = 0.99, Figure 2H). Moreover, the analysis showed that male GEPR-3s have a lower preference for 30% ethanol than male SD rats (q = 5.25, *p* = 0.003, Figure 2H); no statistical differences were observed between the females of the two strains. For water intake, two-way ANOVA revealed significant differences in the sex factor (F_1,44_ = 12.65, *p* = 0.0009) and factor of interaction between strain and sex (F_1,44_ = 6.17, *p* = 0.01) but not in the strain factor (F_1,44_ = 0.40, *p* = 0.527) during the 30% ethanol test. Subsequent analysis revealed that water intake was significantly higher in male GEPR-3s than in female GEPR-3s (q = 6.04, *p* = 0.0006, Figure 2I), while no changes were found in SD rats. No statistical differences were found in water intake when comparing male or female GEPR-3s to male or female SD rats. We also found no statistical differences in the weights of GEPR-3s or SD rats during the voluntary ethanol consumption paradigm (see Appendix A).

### 3.5. Effects of Withdrawal from Voluntary Ethanol Consumption on Acoustically Evoked Seizures in GEPR-3s and SD Rats

We evaluated the impact of 24 h withdrawal from voluntary ethanol consumption on acoustically evoked seizure susceptibility in male and female GEPR-3s or male and female SD rats; we compared the results with the group that consumed water instead of ethanol. The Fisher Exact test revealed no statistical differences in the incidences of WRSs (Figure 3A), GTCSs (Figure 3B), and FLE (Figure 3C) following withdrawal from voluntary ethanol consumption in the GEPR-3s (n = 12) compared to the water consumption GEPR-3 group (n = 12). However, it is worth noting that withdrawal from voluntary consumption slightly increased the occurrence of FLE in 25% of male and 8.5% of female GEPR-3s compared to the water consumption GEPR-3 group (Figure 2C). In one female GEPR-3, the occurrence of FLE was associated with higher ethanol consumption. However, in three male GEPR-3s, the occurrence of FLE was associated with low alcohol intake. No acoustically evoked seizures were observed in male (n=12) and female (n=12) SD rats following withdrawal from voluntary ethanol drinking or water consumption (WRS incidence: 0%; GTCS incidence: 0%, and FLE incidence: 0%). Under control conditions (voluntary water consumption group), the seizure latencies were 15.67 ± 1.22 s (n = 12) and 13.08 ± 0.58 s (n = 12) in male and female GEPR-3s, respectively (Figure 3D); the seizure durations were 31.42 ± 2.09 s (n = 12) and 30.5 ± 2.14 s (n = 12) in male and female GEPR-3s, respectively (Figure 3E). In the voluntary ethanol consumption group, the seizure latencies were 9 ± 1.27 s (n = 12) and 8.92 ± 1.43 s (n = 12) in male and female GEPR-3s, respectively (Figure 3D); the seizure durations were 46.17 ± 4.03 s (n = 12) and 42.83 ± 3.91 s (n = 12) in male and female GEPR-3s, respectively (Figure 3E). Two-way ANOVA revealed a significant difference in the seizure latencies (F_3,44_ = 5.12, *p* = 0.0004) following withdrawal from voluntary ethanol consumption. Furthermore, the seizure latency was reduced in male GEPR-3s subjected to withdrawal from voluntary ethanol consumption compared to the water consumption GEPR-3 group (q = 4.57, *p* < 0.012, Figure 3E). The analysis also revealed that withdrawal from voluntary ethanol consumption significantly altered the seizure duration (F_3,44_ = 6.19, *p* = 0.0013). Multiple comparisons revealed that withdrawal from voluntary ethanol consumption significantly increased seizure duration in male GEPR-3s compared to water consumption male GEPR-3 group (q = 4.64, *p* = 0.010, Figure 3E) or in female GEPR-3s compared to the water consumption female GEPR-3 group (q = 3.83, *p* = 0.044, Figure 3E). We also examined whether withdrawal from voluntary ethanol consumption had any impact on the severity of acoustically evoked seizures. The Kruskal–Wallis test revealed no statistical differences in the seizure severity following withdrawal from voluntary ethanol consumption in GEPR-3s compared to the water consumption GEPR-3s (H = 1.06, *p* = 0.78, Figure 3F).

### 3.6. Acoustically Evoked Seizure Susceptibility at Postnatal Day 28 in Seizures in GEPR-3s and SD Rats

Last, we determined whether P28 GEPR-3s had inherited seizure susceptibility that persisted into adulthood (P112 GEPR-3s). The GEPR-3s and SD rats were subjected to three tests for acoustically evoked seizures between P21 and P28. All tested P28 GEPR-3s (12 males and 12 females) experienced WRSs (Figure 4A) that progressed into GTCSs (Figure 4B). No FLE was observed; therefore, the median seizure severity score was 3 (Figure 4F). No statistical differences in seizure latency and duration were found between male P28 GEPR-3s (seizure latency 19 ± 5.8 and seizure duration 56.5 ± 5.7, n = 12, Figure 4D,E) and female P28 GEPR-3s (seizure latency 16 ± 3.3 and seizure duration 53.6 ± 9.44, n = 12, Figure 4D,E). No acoustically evoked seizure susceptibility was observed in male (n = 12) and female (n = 12) P21-P28 SD rats (WRS incidence: 0%; GTCS incidence: 0%, and FLE incidence: 0%). For the seizure latency, two-way ANOVA comparing male (n = 12) and female (n = 12) P28 GEPR-3s with male (n = 12) and female (n = 12) P112 GEPR-3s revealed significant differences in the sex factor (F_1,44_ = 4.19, *p* = 0.04) and age factor (F_1,44_ = 5.18, *p* = 0.02), but not in the interaction of these factors (F_1,44_ = 0.08, *p* = 0.77). Multiple comparisons revealed no statistical differences in the seizure latency in male and female P28 GEPR-3s compared to male and female P112 GEPR-3s (compare Figure 3D and Figure 4D). For the seizure duration, two-way ANOVA comparing male (n = 12) and female (n = 12) P28 GEPR-3s with male (n = 12) and female (n = 12) P112 GEPR-3s revealed significant differences in the age factor (F_1,44_ = 118.8, *p* < 0.0001), but not in the sex factor (F1,44 = 0.52, *p* = 0.47) or the interaction of these factors (F_1,44_ = 0.11, *p* = 0.73). Multiple comparisons revealed that the seizure duration was significantly decreased in male P112 GEPR-3s compared to male P28 GEPR3-s (compare Figure 3E and Figure 4E) and in female P112 GEPR-3s compared to female P28 GEPR-3s (compare Figure 3E and Figure 4E). No statistical differences were found in the seizure severity between male and female P28 GEPR-3s compared to male and female P112 GEPR-3s (compare Figure 3F and Figure 4F).

## 4. Discussion

The present study delved into the potential link between genetic seizure susceptibility and a predisposition to voluntary alcohol consumption and how withdrawal from voluntary ethanol consumption alters inherited seizure susceptibility in a preclinical model of generalized tonic–clonic epilepsy. Our data indicate that male and female epileptic GEPR-3s consumed less alcohol and had a lower alcohol preference than male and female non-epileptic SD rats, particularly at relatively lower concentrations. However, female GEPR-3s consumed more alcohol and had higher alcohol preferences than male GEPR-3s, while female SD rats consumed more alcohol than male SD rats, but no differences were observed in the alcohol preference. Our findings also revealed that the seizure susceptibility seen at P28 persisted in adult rats, which exhibited short seizure duration. Moreover, withdrawal from chronic voluntary alcohol consumption facilitated the seizure onset and increased the seizure duration in adult GEPR-3s without affecting the seizure severity. Overall, these findings suggest that the genetic seizure susceptibility in GEPR-3s is negatively linked with voluntary alcohol consumption. However, withdrawal from low to moderate amounts of alcohol intake in the epileptic GEPR-3s can promote epileptogenesis. It is, therefore, tempting to speculate that inherited generalized tonic–clonic epilepsy does not predispose to higher alcohol consumption and preference and that patients with this type of epilepsy should refrain from drinking even lower amounts of alcohol.

Clinical studies have provided conflicting information on the link between epilepsy and alcohol consumption [4,5,6,7,25]. Our findings are consistent with studies reporting that mice with higher susceptibility to AWSs tend to consume less alcohol than those with lower susceptibility [11,12]. Another report indicated that alcohol consumption and subsequent withdrawal can transiently increase the incidence of spontaneous recurrent seizures in the pilocarpine post-status epilepticus model [10]. Withdrawal from high doses of alcohol administration can increase the duration of postictal depression and respiratory distress and trigger the occurrence of post-tonic generalized clonus in GEPR-9s [26]. In contrast, our findings demonstrated that withdrawal from voluntary alcohol consumption only slightly increased the occurrence of partial tonic seizures and facilitated alcohol withdrawal-induced epileptogenesis in GEPR-3s. The discrepancies in the development of AWSs in the pilocarpine and GEPR models may lie in the network for seizures. In the pilocarpine post-status epilepticus model, limbic seizures originate from the hippocampus, while brainstem seizures in GEPR-3s are triggered in the inferior colliculus; both these seizure types are not directly linked to withdrawal from increased alcohol intake [15,27]. Repetition of acoustically evoked seizures (audiogenic kindling) in GEPR-3s can result in the occurrence of brainstem-triggered limbic seizures similar to those seen in the pilocarpine model [28]. Interestingly, evidence indicated that inhibiting neuronal activity in a circuit between the premedial frontal cortex and dorsal periaqueductal, which might be affected by audiogenic kindling, can promote compulsive alcohol consumption [29]. It is plausible that GEPR-3s may consume more alcohol and exhibit higher incidences of AWSs following audiogenic kindling. Our findings on lower alcohol intake in GEPR-3s are also consistent with data obtained using non-epileptic mutant SD rats, which exhibit lower alcohol intake [20,21]. These findings suggest that the aversive effect of alcohol may depend on mechanisms underlying genetic variations [30,31,32,33,34]. It is also plausible that low voluntary alcohol consumption in GEPR-3s may be due to adverse motivational effects or taste aversion; this protective mechanism may reduce the probability of developing a pattern of excessive alcohol consumption [35,36]. Interestingly, the inferior colliculus, the origin site of acoustically evoked seizures, can respond to rewarding and aversive stimuli [15,37,38,39].

The mechanisms underlying the negative correlation between inherited seizure susceptibility and voluntary alcohol intake in GEPR-3s are not fully understood. However, an analysis of the quantitative trait locus related to AWSs has identified specific genes that might play important roles in AWSs, including genes that encode for potassium channels (Kcnma1, Kcnj6), sodium channels (Scn8a), ryanodine receptors (Ryr1, Ryr2), gamma-aminobutyric acid receptors (Gabra4), and glutamate receptors (Grin2a, Grin2b) [40]. These genes may also reduce alcohol consumption associated with increased AWS susceptibility [41,42]. Interestingly, studies reported activating the small conductance calcium-activated potassium channels reduced rodent alcohol consumption [43,44,45]. Additionally, retigabine, a Kv7 channel opener, reduced rodent alcohol consumption [42,46]. Furthermore, mice lacking Girk3, one of the four G protein-coupled inwardly rectifying potassium channel subunits, exhibited excessive alcohol drinking [47]. Likewise, altered gene expression in specific brain areas may also modulate alcohol consumption. A report indicated that genetic knockdown of hyperpolarization-activated cyclic nucleotide-gated ion channel type 2 in the ventral tegmental area reduces alcohol consumption in alcohol-preferring rats [48]. Furthermore, focal microinjections of dopamine or glutamate receptor antagonists within the nucleus accumbens also reduced ethanol self-administration [49]. A study reported low levels of expression of dopamine transporter mRNA in the ventral tegmental area of GEPR-3s, suggesting changes in dopamine’s effects in the ventral tegmental area projection’s terminal regions, including the nucleus accumbens [50]. In other models of inherited susceptibility to acoustically evoked generalized seizures, reduced dopamine D1-like and D2-like receptor binding densities have been observed in the nucleus accumbens [51]. The downregulation of D1-like and D2-like receptors may also occur in the nucleus accumbens of GEPR-3s and contribute to lower alcohol intake. Moreover, evidence indicates that innate high levels of anxiety motivate excessive alcohol consumption [52]. Intriguingly, GEPR-3s showed high inherited anxiety but consumed less alcohol when compared to non-seizure-susceptible SD rats [53].

In this study, we also measured water intake to determine if changes in the levels of alcohol intake were associated with water consumption. Our findings indicate that female GEPR-3s and SD rats consumed more alcohol when compared to males, without any significant increase in water intake. These findings align with previous studies on alcohol consumption and sex differences [54,55]. However, the underlying mechanisms of this trend are not fully understood. Higher voluntary alcohol intake in females may be linked to the estrus cycle and circulating hormones, which were not assessed in this study [56]. Nevertheless, reports suggested that estradiol, an ovarian hormone, contributed to increased alcohol consumption in ovariectomized female mice; however, ethanol consumption was not affected by the estrous cycle phase in freely cycling female mice [57,58,59,60]. These findings suggest that females may find ethanol more rewarding than males because of higher circulating estradiol levels [61]. Additionally, the pharmacokinetics of alcohol may also play a role in alcohol consumption and sex differences, as females may need to consume more alcohol to achieve higher blood alcohol levels [62].

One limitation of our study is that rats were housed in single cages, and this social isolation may contribute to low alcohol consumption in GEPR-3s. However, there are conflicting reports on the impact of social isolation on alcohol intake. A study reported that both young and adult rats exhibited increased alcohol intake when socially isolated [63]. Furthermore, female rats are more sensitive to social isolation, which would affect their higher alcohol consumption when compared to males [64]. In contrast, another study revealed that male rats housed in pairs had higher alcohol intake preference compared to those housed individually; no difference was observed in alcohol intake and preference among females based on housing conditions [65].

In conclusion, we observed a negative association between hereditary seizure susceptibility and voluntary alcohol consumption and preference for alcohol, indicating the need for further studies to assess the mechanisms underlying this discrepancy.

## Figures and Tables

**Figure 1 brainsci-14-00188-f001:**
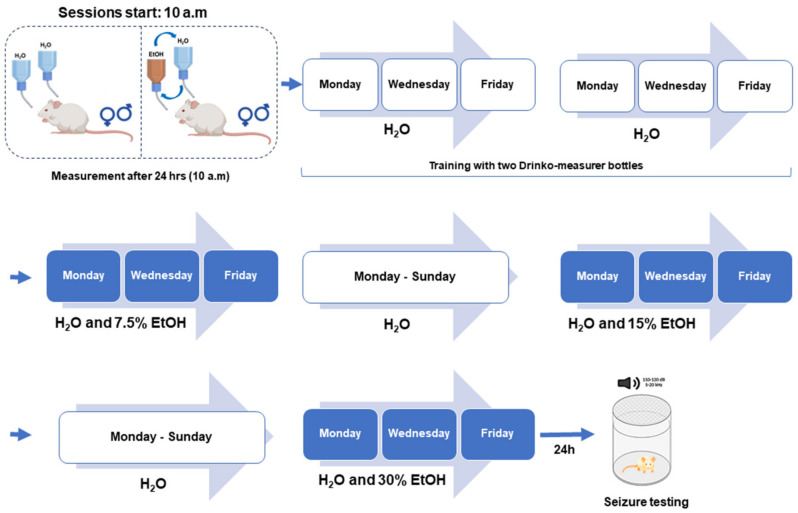
Experimental design. For voluntary ethanol (EtOH) consumption, twenty-four seizure-susceptible genetically epilepsy-prone rats (GEPR-3s; 12 males and 12 females) and twenty-four Sprague Dawley (SD) rats (12 males, 12 females) were used. The rats were first acclimated to drinking from two bottles containing water. Subsequently, they were given access to two bottles simultaneously, one containing water and the other 7.5%, 15%, or 30% (*v*/*v*) alcohol three times a week, followed by one week of only water for each alcohol dose. For acoustically evoked seizure testing, GEPR-3s (12 males and 12 females) previously subjected to voluntary ethanol and GEPR-3s (12 males and 12 females) subjected to voluntary water drinking were used. All animals were tested for acoustically evoked seizures between the 20th and 24th hour after removing ethanol- or water-containing bottles.

**Figure 2 brainsci-14-00188-f002:**
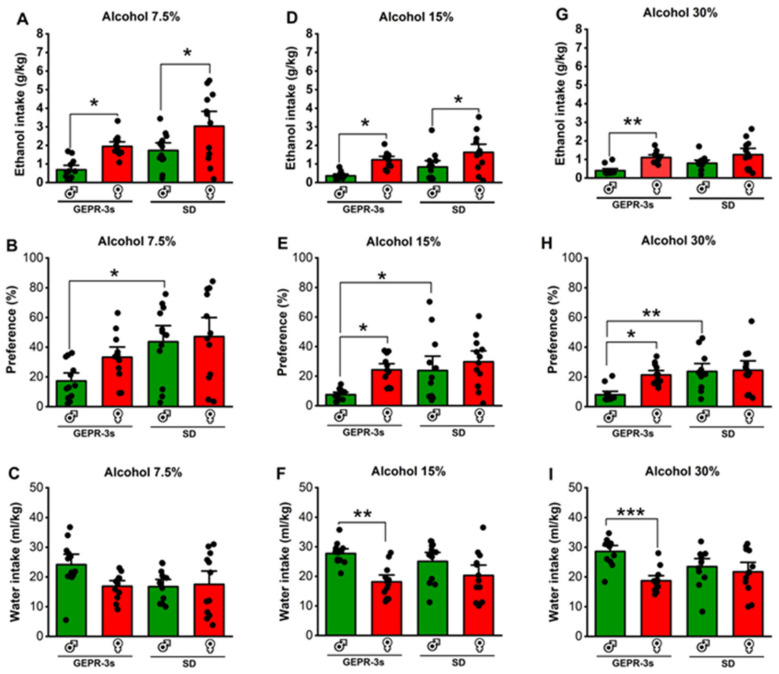
Effects of sex and strain on ethanol intake (g/kg), preference (%), and water (mL/kg). Male and female genetically epilepsy-prone rats (GEPR-3s) and Sprague Dawley (SD) rats were exposed to intermittent voluntary ethanol consumption, as described in Figure 1. (**A**) Female GEPR-3s and SD rats consumed more 7.5% ethanol than males. (**B**) Male SD rats exhibited a higher preference for 7.5% ethanol compared to male GEPR-3s. (**C**) No statistical differences were observed in water consumption between GEPR-3s and SD rats during the 7.5% ethanol test. (**D**) Female GEPR-3 and SD rats consumed more 15% ethanol than males. (**E**) Female GEPR-3s exhibited a higher preference for 15% ethanol than males. (**F**) Male GEPR-3s consumed more water than females during the 15%. (**G**) Only female GEPR-3 consumed more 30% ethanol than males. (**H**) Female GEPR-3s exhibited a higher preference for 30% ethanol than males. Furthermore, male SD rats exhibited a higher preference for 30% ethanol than male GEPR-3s. (**I**) Male GEPR-3s consumed more water than females during the 30%. Data are presented as percentage (%) for preference and mean ± S.E.M. for alcohol intake (g/kg) and water intake (mL/kg). The data were analyzed using two-way ANOVA repeated measures followed by Tukey post hoc correction (* *p* < 0.05, ** *p* < 0.01, and *** *p* < 0.001, n = 12).

**Figure 3 brainsci-14-00188-f003:**
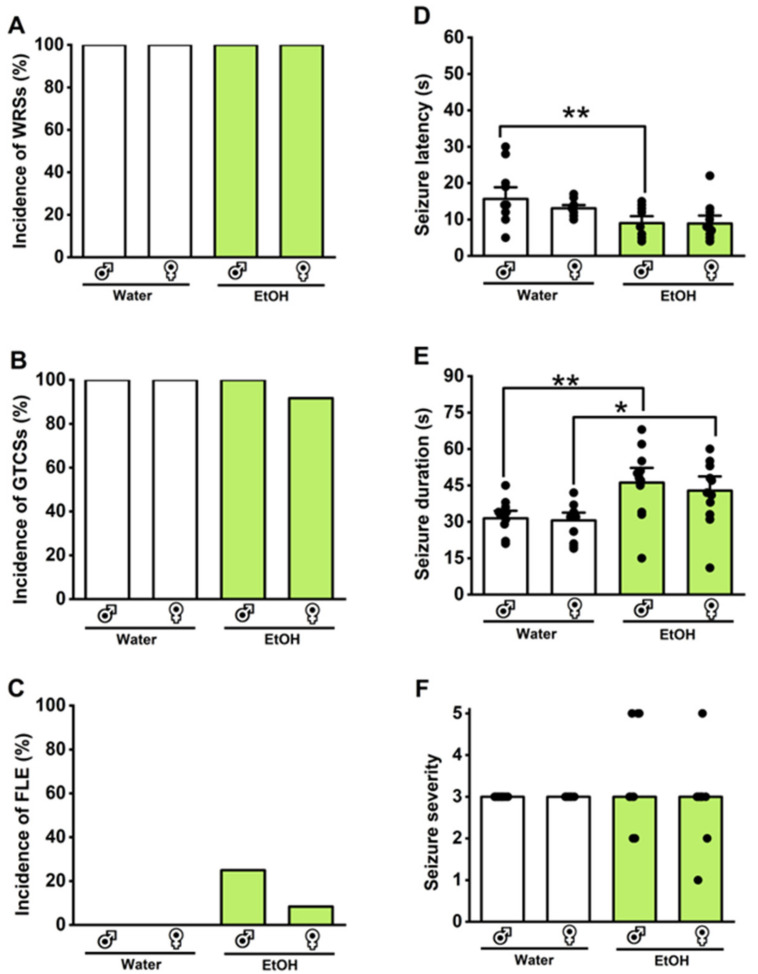
Effects of withdrawal from voluntary ethanol consumption on the acoustically evoked seizure susceptibility in the genetically epilepsy-prone rats (GEPR-3s). Male and female GEPR-3s were subjected to intermittent voluntary ethanol consumption or water, as illustrated in Figure 1. During withdrawal from voluntary ethanol consumption, there were no noticeable changes in the incidences of wild running seizures (WRSs, (**A**)), generalized tonic–clonic seizures (GTCSs, (**B**)), and forelimb extension (FLE, (**C**)). However, statistical differences were observed in the reduced seizure latency, defined as the onset of the first WRSs episode (**D**), and an increased total seizure (WRSs and GTCSs) duration (**E**) in both male and female GEPR-3s. There were no statistical differences in the seizure severity of all seizure phenotypes (**F**). Data are presented as percentage (%) for the incidences of WRSs, GTCSs, and FLE, analyzed using the Fischer test; mean ± S.E.M. for seizure latency and duration, analyzed using one-way ANOVA followed by Tukey post hoc correction; and median score ± median average deviation for the seizure severity, analyzed using Kruskal–Wallis test (* *p* < 0.05, ** *p* < 0.01; n = 12).

**Figure 4 brainsci-14-00188-f004:**
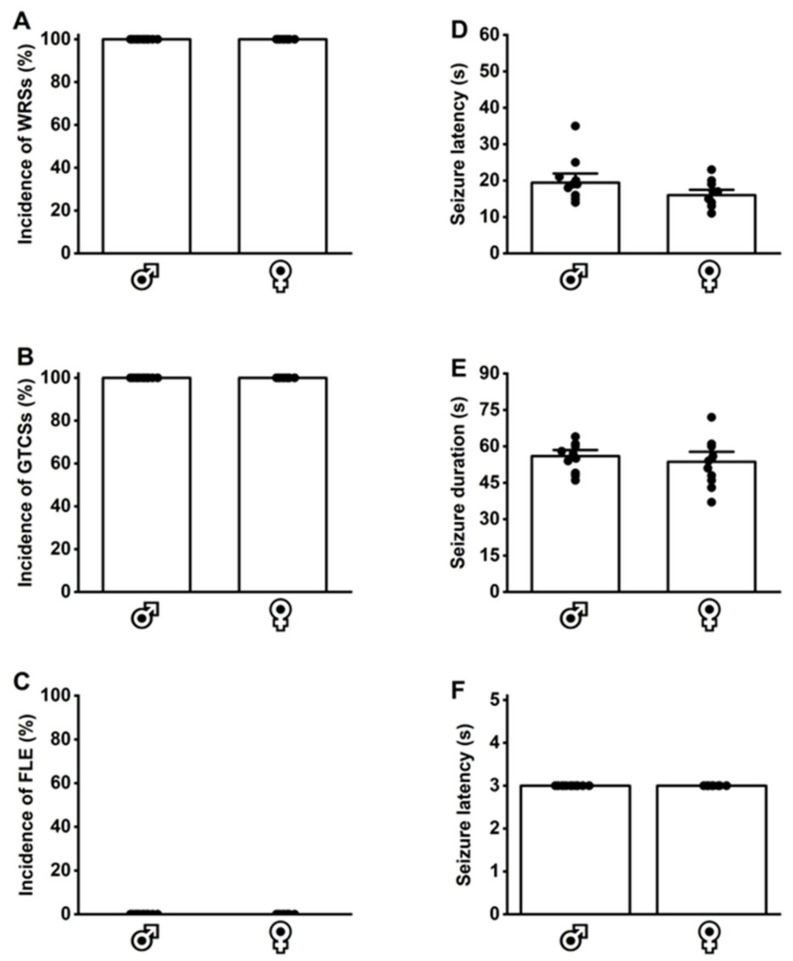
Inherited seizure susceptibility in postnatal day 28 genetically epilepsy-prone rats (GEPR-3s). Male (n = 12) and female (n = 12) GEPR-3s underwent acoustically evoked seizure testing. (**A**) All GEPR-3s (male and female) experienced wild running seizures (WRSs) and (**B**) generalized tonic–clonic seizures (GTCSs). (**C**) However, no forelimb extension (FLE) was observed in male and female GEPR-3s. (**D**,**E**) No statistical differences in seizure latency and duration were observed between male and female GEPR-3s. (**F**) All GEPR-3s experienced seizure severity stage 3. Data are presented as percentage (%) for the incidences of WRSs, GTCSs, and FLE, analyzed using the Fischer test; mean ± S.E.M. for seizure latency and duration, analyzed using one-way ANOVA followed by Tukey post hoc correction; and median score ± median average deviation for the seizure severity, analyzed using the Mann–Whitney test.

## Data Availability

The data presented in this study are available on request from the corresponding author.

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
