# Peer review of "Influence of Inherited Seizure Susceptibility on Intermittent Voluntary Alcohol Consumption and Alcohol Withdrawal Seizures in Genetically Epilepsy-Prone Rats (GEPR-3s)"

_brainsci, 2024, doi:10.3390/brainsci14020188_

Round 1
Reviewer 1 Report
Comments and Suggestions for Authors
The current study examined the impact of alcohol consumption and alcohol preference on seizure-prone GEPR-3s rats (12 male and 12 female subjects) and control Sprague Dowley rats (12 males and 12 females). This paper is well written and well illustrated. I have some questions about fundamental concepts and hypotheses.
The aim of the current study should be clearly defined and specific.
Lines 65-67. "Our study aims to investigate the influence of inherited seizure susceptibility on short-term voluntary alcohol consumption and AWS susceptibility in male and female GEPR-3s compared to SD rats, which exhibit no seizure susceptibility." This might suggest the following hypotheses, (1) inherited seizure susceptibility influences short-term voluntary alcohol consumption; (2) there are sex differences in susceptibility to alcohol withdrawal seizures. How were alcohol withdrawal seizures in GEPR-3s rats and control SD rats assessed?
How old were the rats in the core experimental group, including 24 GEPR-3s and 24 SD rats?
The voluntary consumption of alcohol is an important issue as opposed to the commonly used method of intragastric administration. I have a question about the experimental design, which is not clear from Figure 1. How many days did rats consume 7.5% alcohol? Were there only three 24-hour sessions of alcohol consumption on Monday, Wednesday, and Friday? After one week of withdrawal, they were presented with 15% alcohol, and one week later they were presented with 30% alcohol. Please clarify this issue in Method section.
The authors mentioned alcohol withdrawal seizures. Please be aware that alcohol withdrawal seizures are spontaneous, and they are a severe manifestation of alcohol withdrawal syndrome, occurring 6-48 hours after alcohol cessation as blood alcohol returned to zero. I think that "alcohol withdrawal seizures" are misleading here. Lines 59-61: "The GEPR-3s were explicitly included in this study because they exhibit GTCSs *generalized tonic-clonic seizures* that are similar to the most common type of alcohol withdrawal seizures [18,19]." These references do not seem to support this point.
18.Mishra et al (1989) doi:10.1016/0920-1211(89)90023-5. This paper did not discuss alcohol withdrawal seizures and mentioned that "GEPR-3s are selectively bred for class 3 seizures".
19.Faingold (2008) doi:10.1002/0471142301.ns0928s44. This paper describes behavioral signs of intoxication after investigator-initiated intragastric ad- ministration of 5 g/kg ethanol over a period of 4 days. Audiogenic seizures are a part of ethanol withdrawal behavior that could be seen in control rats.
Line 94. "The body weight of each rat was measured daily". Line 268-269. "no statistical differences in the weights of GEPR-3s or SD rats during the voluntary ethanol consumption paradigm (data not shown)." The overall tests lasted 7 weeks (as follows from Fig. 1), and alcohol consumption on the 5-th and 7-th weeks implies intoxication and changes in body weight. Changes in body weight should be related to intoxication degree and further related to severity of audiogenic seizures. Can the authors provide more specific information regarding the issue of body weight instead of simply stating "data not shown"?
The study of Faingold (2008) assumes that alcohol consumption might induce audiogenic seizures in SD rats. But the authors reported: "No acoustically evoked seizures were observed in male and female SD rats following withdrawal from voluntary ethanol drinking or water consumption (data not shown)." SD rats did not show any signs of audiogenic seizures, suggesting that they might not be affected by intoxication. Considering that alcohol consumption in GEPR-3s and SD rats was the same, GEPR-3s might not be affected by intoxication. Therefore, the concept of alcohol withdrawal seizures is misleading.
The authors classified severity of audiogenic seizures based on Mishra et al's (1989) creteria, namely
WRS - wild running seizures (mild form of audiogenic epilepsy, audiogenic re- sponse score of 1).
GTCSs - generalized tonic-clonic seizures (middle severity of audiogenic epilepsy, class 3, typical for GEPR-3s strain).
FLE - forelimb extension (severe audiogenic epilepsy, class >3).
Figure 3F shows that ethanol consumption in GEPR-3s rats affected seizure severity: the severity score was equal to 3 in all rats from the water group, and varied from 1 to 5 in the ethanol group. How to explain this individual variability? Perhaps severity scores decreased in rats with a high ethanol preference?
Considering that the vast majority of GEPR-3s exhibit generalized tonic-clonus seizures (class 3) [Mishra et al (1989) doi:10.1016/0920-1211(89)90023-5], external factors, such as alcohol intake, increased susceptibility to audiogenic seizures only in genetically prone rats.
Minor.
Figure 3. Please correct the legend. С shows seizure latency (all types of seizures?). E shows seizure duration (again, all types of seizures?). F shows seizure severity (all seizure types?)
Author Response
Prof. Dr. Stephen D. Meriney
Editor-in-Chief,
Brain Sciences
RE: brainsci-2873060
Dr. Meriney
I have received reviews for the above-referenced manuscript. We are grateful to the referees for their valuable feedback and constructive comments. After carefully considering their suggestions, we have made significant revisions and clarifications as requested. We believe that the manuscript is now suitable for publication in Brain Sciences. All the changes made in the revisions and our responses to the referees are listed in detail below.
Revisions made by the authors
- We used the “Track changes” option for changes made in the manuscript.
- As suggested by Reviewer 2, we have excluded the following acronyms: WSP, WSF, IC, Nac, VTA, DR1R, and DR2R.
- We insert Supplementary Table 1 in the Results section (Line 281).
- We have included two new references (Lines 551, 637).
- We have made additional changes for enhanced readability and clarity: Lines 67-69, 86-87, 106, 108, 111-116, 186-187, 402-405.
Responses to reviews
Review 1
General comments
The current study examined the impact of alcohol consumption and alcohol preference on seizure-prone GEPR-3s rats (12 male and 12 female subjects) and control Sprague Dowley rats (12 males and 12 females). This paper is well written and well illustrated. I have some questions about fundamental concepts and hypotheses.
RE: Thank you for your positive comments on our manuscript. The questions are relevant, and the suggested modifications can improve our revised version.
Point 1: The aim of the current study should be clearly defined and specific.
Lines 65-67. "Our study aims to investigate the influence of inherited seizure susceptibility on short-term voluntary alcohol consumption and AWS susceptibility in male and female GEPR-3s compared to SD rats, which exhibit no seizure susceptibility." This might suggest the following hypotheses, (1) inherited seizure susceptibility influences short-term voluntary alcohol consumption; (2) there are sex differences in susceptibility to alcohol withdrawal seizures. How were alcohol withdrawal seizures in GEPR-3s rats and control SD rats assessed?
RE: We have three working hypotheses are, Firstly, inherited seizure susceptibility might influence short-term voluntary alcohol consumption in GEPR-3s. Secondly, there might be differences in alcohol consumption between male and female GEPR-3s. And thirdly, alcohol withdrawal could worsen inherited seizure susceptibility in GEPR-3s.
To evaluate alcohol withdrawal seizures in GEPR-3 and control SD rats, the animals were tested for acoustically evoked seizures 24 hours after voluntary alcohol consumption.
Point 2: How old were the rats in the core experimental group, including 24 GEPR-3s and 24 SD rats?
RE: 8-week-old (see Line 73).
Point 3: The voluntary consumption of alcohol is an important issue as opposed to the commonly used method of intragastric administration. I have a question about the experimental design, which is not clear from Figure 1. How many days did rats consume 7.5% alcohol? Were there only three 24-hour sessions of alcohol consumption on Monday, Wednesday, and Friday? After one week of withdrawal, they were presented with 15% alcohol, and one week later they were presented with 30% alcohol. Please clarify this issue in the Method section.
RE: The rats were given 7.5% alcohol to consume on Monday, Wednesday, and Friday, each for 24 hours. After a week of not consuming alcohol, they were given 15% alcohol, and after another week of abstinence, they were given 30% alcohol. After 24 hours of not drinking alcohol, the rats were tested for acoustically evoked seizures (see lines 106-116).
Point 4: The authors mentioned alcohol withdrawal seizures. Please be aware that alcohol withdrawal seizures are spontaneous, and they are a severe manifestation of alcohol withdrawal syndrome, occurring 6-48 hours after alcohol cessation as blood alcohol returned to zero. I think that "alcohol withdrawal seizures" are misleading here. Lines 59-61: "The GEPR-3s were explicitly included in this study because they exhibit GTCSs *generalized tonic-clonic seizures* that are similar to the most common type of alcohol withdrawal seizures [18,19]." These references do not seem to support this point.
18.Mishra et al (1989) doi:10.1016/0920-1211(89)90023-5. This paper did not discuss alcohol withdrawal seizures and mentioned that "GEPR-3s are selectively bred for class 3 seizures".
19.Faingold (2008) doi:10.1002/0471142301.ns0928s44. This paper describes behavioral signs of intoxication after investigator-initiated intragastric administration of 5 g/kg ethanol over a period of 4 days. Audiogenic seizures are a part of ethanol withdrawal behavior that could be seen in control rats.
RE: Thank you for your feedback. We understand that while humans may experience spontaneous seizures accompanied by auditory hallucinations during alcohol withdrawal, such seizures are rare in rodents. However, we have developed a reliable rodent model of alcohol withdrawal where acoustically evoked seizures can be produced. (doi: 10.3390/brainsci11020279; 10.1016/j.alcohol.2017.07.007; 10.1111/acer.13223; 10.1093/ijnp/pyu123; doi.org/10.1002/0471142301.ns0928s44; 10.1016/j.yebeh.2016.09.024). Interestingly, in this model, the phenotype of acoustically evoked generalized tonic-clonic seizures in Sprague-Dawley rats following withdrawal from severe alcohol intoxication is similar to acoustically evoked tonic-clonic seizures in the GEPR-3s but not the GEPR-9s(Mishra et al., 1989). We hope this clarification resolves any issues (Lines 61-62).
Point 5: Line 94. "The body weight of each rat was measured daily". Line 268-269. "no statistical differences in the weights of GEPR-3s or SD rats during the voluntary ethanol consumption paradigm (data not shown)." The overall tests lasted 7 weeks (as follows from Fig. 1), and alcohol consumption on the 5-th and 7-th weeks implies intoxication and changes in body weight. Changes in body weight should be related to intoxication degree and further related to severity of audiogenic seizures. Can the authors provide more specific information regarding the issue of body weight instead of simply stating "data not shown"?
RE: Thanks for your comment. We used the animals' weight to calculate their alcohol intake. Although no statistical differences were found between the two strains, we have included the animal weights in the Supplementary material (Line 281).
Point 6: The study of Faingold (2008) assumes that alcohol consumption might induce audiogenic seizures in SD rats. But the authors reported: "No acoustically evoked seizures were observed in male and female SD rats following withdrawal from voluntary ethanol drinking or water consumption (data not shown)." SD rats did not show any signs of audiogenic seizures, suggesting that they might not be affected by intoxication. Considering that alcohol consumption in GEPR-3s and SD rats was the same, GEPR-3s might not be affected by intoxication. Therefore, the concept of alcohol withdrawal seizures is misleading.
RE: In the Majchrowicz binge alcohol protocol (Faingold, 2008), rats were given high doses of alcohol for several days and then tested for acoustically evoked seizures during alcohol withdrawal. Although this method is not directly applicable to humans, it has been linked with an increased incidence of seizures. In our study, we found that both SD rats and GEPR-3s voluntarily consumed low amounts of alcohol but that GEPR-3s were more susceptible to acoustically evoked seizures during the withdrawal than SD rats. Specifically, we observed that GEPR-3s had a decreased seizure latency and an increased seizure duration, indicating that they were experiencing alcohol withdrawal-induced hyperexcitability. This suggests that the concept of alcohol withdrawal seizures is relevant to our study.
Point 7: The authors classified severity of audiogenic seizures based on Mishra et al's (1989) criteria, namely WRS - wild running seizures (mild form of audiogenic epilepsy, audiogenic response score of 1).
GTCSs - generalized tonic-clonic seizures (middle severity of audiogenic epilepsy, class 3, typical for GEPR-3s strain). FLE - forelimb extension (severe audiogenic epilepsy, class >3).
Figure 3F shows that ethanol consumption in GEPR-3s rats affected seizure severity: the severity score was equal to 3 in all rats from the water group, and varied from 1 to 5 in the ethanol group. How to explain this individual variability? Perhaps severity scores decreased in rats with a high ethanol preference?
RE: Thank you for your comments. We have found that voluntary ethanol consumption is linked to a slight increase in FLE incidence in GEPR-3s. Alcohol preference was higher in one female GEPR-3 that had FLE and reduced seizure latency, while two female GEPR-3s had lower alcohol preference and reduced seizure latency and duration. On the other hand, three male GEPR-3s had FLE associated with increased seizure duration, whereas three others had a seizure score of 2 and reduced seizure duration despite lower alcohol preference (Lines 294-297). These observations demonstrate biological variability in ethanol intake and seizure susceptibility.
Point 8: Considering that the vast majority of GEPR-3s exhibit generalized tonic-clonus seizures (class 3) [Mishra et al (1989) doi:10.1016/0920-1211(89)90023-5], external factors, such as alcohol intake, increased susceptibility to audiogenic seizures only in genetically prone rats.
RE: Thank you for your feedback. Low alcohol consumption can reduce the onset of seizures and prolong their duration, leading to the emergence of FLE only in the seizure-prone GEPR-3s. Alcohol consumption may increase the risk of acoustically evoked seizures in GEPR-3s. However, inherited seizure susceptibility alone does not necessarily increase alcohol consumption compared to non-seizure-prone SD rats.
Minor.
Point 9: Figure 3. Please correct the legend. С shows seizure latency (all types of seizures?). E shows seizure duration (again, all types of seizures?). F shows seizure severity (all seizure types?)
RE: Thank you for your feedback. Based on your comments, we have made the necessary changes (Lines 326-331).
Reviewer 2 Report
Comments and Suggestions for Authors
Title: Influence of inherited seizure susceptibility on intermittent
voluntary alcohol consumption and alcohol withdrawal seizures in genetically
epilepsy-prone rats (GEPR-3s)
Authors: Gleice Kelli Silva-Cardoso, Prosper N'Gouemo *
Neuropharmacology and Neuropathology
The paper addresses the differences in voluntary alcohol consumption and withdrawal seizures in a mutant rat model, the genetically epilepsy-prone rat (GEPR-3s), and a control strain, the non-mutant sprague dawley rat. The association between genetic epilepsy and alcohol use and susceptibility is not well understood In addition to Strain, Sex and Age differences were also investigated. Overall, GEPR-3 seizure susceptible rats drank less alcohol than the control rats. When they did drink, GEPR-3 seizure outcomes worsened. Females drank more than males in both strains.
The paper is well written and the data supports the conclusions. The hypotheses are clear and investigated in an appropriate way. Only minor edits need to be made before publication:
As a general comment, there are many acronyms used throughout the paper and it can be a bit confusing for the reader. I would suggest limiting the use of acronyms to the most frequently used terms only.
Subjects: Please provide the total number of animals used in the study (is it 96?). Since you bred the animals for seizure susceptibility and also suggest generalized mutation effects on alcohol use I think it is important to include details about the parent generation and early offspring environment. What generation are these offspring? How are the parents paired? Do both male and female parents have seizures that are characterized? How are offspring caged at weaning/ when did you isolate them?
Methods: what is the purpose of the one week of water in between the drinking dilutions? It doesn’t seem very translational. Would these wash out periods affect subsequent withdrawal seizures?
Please check line 145-146 for grammar affecting clarity.
Please provide a rationale or reference for preference of using a q-value.
Please check lines 173-176 for clarity.
Fig. 3 Please check that the figures are in the correct order. E.g., line 314 has WRSs as D
Line 392: IC is not defined, (move up from line 406)
Please provide a reference for line 447.
Abstract: line 22: did you run a correlation analysis? If not, remove ‘correlated’

Author Response
Prof. Dr. Stephen D. Meriney
Editor-in-Chief,
Brain Sciences
RE: brainsci-2873060
Dr. Meriney
I have received reviews for the above-referenced manuscript. We are grateful to the referees for their valuable feedback and constructive comments. After carefully considering their suggestions, we have made significant revisions and clarifications as requested. We believe that the manuscript is now suitable for publication in Brain Sciences. All the changes made in the revisions and our responses to the referees are listed in detail below.
Revisions made by the authors
- We used the “Track changes” option for changes made in the manuscript.
- As suggested by Reviewer 2, we have excluded the following acronyms: WSP, WSF, IC, Nac, VTA, DR1R, and DR2R.
- We insert Supplementary Table 1 in the Results section (Line 281).
- We have included two new references (Lines 551, 637).
- We have made additional changes for enhanced readability and clarity: Lines 67-69, 86-87, 106, 108, 111-116, 186-187, 402-405.
Responses to reviews
Review 2
General comments
Point 1: The paper addresses the differences in voluntary alcohol consumption and withdrawal seizures in a mutant rat model, the genetically epilepsy-prone rat (GEPR-3s), and a control strain, the non-mutant sprague dawley rat. The association between genetic epilepsy and alcohol use and susceptibility is not well understood In addition to Strain, Sex and Age differences were also investigated. Overall, GEPR-3 seizure susceptible rats drank less alcohol than the control rats. When they did drink, GEPR-3 seizure outcomes worsened. Females drank more than males in both strains. The paper is well written and the data supports the conclusions. The hypotheses are clear and investigated in an appropriate way. Only minor edits need to be made before publication:
RE: Thank you for your positive feedback on our manuscript. We believe that your suggested modifications will significantly improve the revised version.
Point 2: As a general comment, there are many acronyms used throughout the paper, and it can be a bit confusing for the reader. I would suggest limiting the use of acronyms to the most frequently used terms only.
RE: "Thank you for your feedback; we have considered it and made the necessary changes.".
Point 3: Subjects: Please provide the total number of animals used in the study (is it 96?). Since you bred the animals for seizure susceptibility and also suggest generalized mutation effects on alcohol use I think it is important to include details about the parent generation and early offspring environment. What generation are these offspring? How are the parents paired? Do both male and female parents have seizures that are characterized? How are offspring caged at weaning/ when did you isolate them?
RE: Thank you for your comments. The total number of animals is 96, as mentioned in Line 73. The offspring GEPR-3s used in this study belong to the 45th generation of our colony (Lines 78-79). During the breeding process, we paired one male and one female in the same cage (Line 80). To ensure the presence of the genetic seizure trait in our colony, we only mate males and females that had generalized tonic-clonic seizures (Lines 80-81). After weaning at postnatal day 21, GEPR-3s were housed in pairs of the same sex per cage, as stated in lines 81-82. Adult animals are separated individually for five days (acclimation) before starting the experiments, as mentioned in Line xx.
Point 4: Methods: what is the purpose of the one week of water in between the drinking dilutions? It doesn’t seem very translational. Would these wash out periods affect subsequent withdrawal seizures?
RE: Thank you for your feedback. To minimize interference when measuring animals' ethanol intake, we implemented a 7-day washout period for different alcohol concentrations. We do not believe that these washout periods would impact subsequent alcohol withdrawal seizures.
Point 5: Please check line 145-146 for grammar affecting clarity.
RE: Thank you for your comments; we have made the necessary changes (Lines 155-157).
Point 6: Please provide a rationale or reference for preference of using a q-value.
RE: The Tukey test compares the difference between each pair of means of the groups with an appropriate adjustment for the multiple testing, called the q critical value (Lines 169-170).
Point 7: Please check lines 173-176 for clarity.
RE: Thank you for your comment; we made changes accordingly (Lines 186-187).
Point 8: Fig. 3 Please check that the figures are in the correct order. E.g., line 314 has WRSs as D
RE: Thank you for your comment. We have made the necessary changes (Lines 327-331).
Point 9: Line 392: IC is not defined, (move up from line 406)
RE: Thank you for your comment; we have made changes accordingly (Line 410).
Point 10: Please provide a reference for line 447.
RE: Thank you for your comment. We have made the necessary changes (Line 467).
Point 11: Abstract: line 22: did you run a correlation analysis? If not, remove ‘correlated’.
RE: Thank you for bringing this to our attention; we have made a change to address your comment (Line 23).
We hope that this manuscript is now suitable for publication in Brain Sciences.
Sincerely,
Prosper N'Gouemo
Associate Professor
prosper.ngouemo@howard.edu
Prof. Dr. Stephen D. Meriney
Editor-in-Chief,
Brain Sciences
RE: brainsci-2873060
Dr. Meriney
I have received reviews for the above-referenced manuscript. We are grateful to the referees for their valuable feedback and constructive comments. After carefully considering their suggestions, we have made significant revisions and clarifications as requested. We believe that the manuscript is now suitable for publication in Brain Sciences. All the changes made in the revisions and our responses to the referees are listed in detail below.
Revisions made by the authors
- We used the “Track changes” option for changes made in the manuscript.
- As suggested by Reviewer 2, we have excluded the following acronyms: WSP, WSF, IC, Nac, VTA, DR1R, and DR2R.
- We insert Supplementary Table 1 in the Results section (Line 281).
- We have included two new references (Lines 551, 637).
- We have made additional changes for enhanced readability and clarity: Lines 67-69, 86-87, 106, 108, 111-116, 186-187, 402-405.
Responses to reviews
Review 2
General comments
Point 1: The paper addresses the differences in voluntary alcohol consumption and withdrawal seizures in a mutant rat model, the genetically epilepsy-prone rat (GEPR-3s), and a control strain, the non-mutant sprague dawley rat. The association between genetic epilepsy and alcohol use and susceptibility is not well understood In addition to Strain, Sex and Age differences were also investigated. Overall, GEPR-3 seizure susceptible rats drank less alcohol than the control rats. When they did drink, GEPR-3 seizure outcomes worsened. Females drank more than males in both strains. The paper is well written and the data supports the conclusions. The hypotheses are clear and investigated in an appropriate way. Only minor edits need to be made before publication:
RE: Thank you for your positive feedback on our manuscript. We believe that your suggested modifications will significantly improve the revised version.
Point 2: As a general comment, there are many acronyms used throughout the paper, and it can be a bit confusing for the reader. I would suggest limiting the use of acronyms to the most frequently used terms only.
RE: "Thank you for your feedback; we have considered it and made the necessary changes.".
Point 3: Subjects: Please provide the total number of animals used in the study (is it 96?). Since you bred the animals for seizure susceptibility and also suggest generalized mutation effects on alcohol use I think it is important to include details about the parent generation and early offspring environment. What generation are these offspring? How are the parents paired? Do both male and female parents have seizures that are characterized? How are offspring caged at weaning/ when did you isolate them?
RE: Thank you for your comments. The total number of animals is 96, as mentioned in Line 73. The offspring GEPR-3s used in this study belong to the 45th generation of our colony (Lines 78-79). During the breeding process, we paired one male and one female in the same cage (Line 80). To ensure the presence of the genetic seizure trait in our colony, we only mate males and females that had generalized tonic-clonic seizures (Lines 80-81). After weaning at postnatal day 21, GEPR-3s were housed in pairs of the same sex per cage, as stated in lines 81-82. Adult animals are separated individually for five days (acclimation) before starting the experiments, as mentioned in Line xx.
Point 4: Methods: what is the purpose of the one week of water in between the drinking dilutions? It doesn’t seem very translational. Would these wash out periods affect subsequent withdrawal seizures?
RE: Thank you for your feedback. To minimize interference when measuring animals' ethanol intake, we implemented a 7-day washout period for different alcohol concentrations. We do not believe that these washout periods would impact subsequent alcohol withdrawal seizures.
Point 5: Please check line 145-146 for grammar affecting clarity.
RE: Thank you for your comments; we have made the necessary changes (Lines 155-157).
Point 6: Please provide a rationale or reference for preference of using a q-value.
RE: The Tukey test compares the difference between each pair of means of the groups with an appropriate adjustment for the multiple testing, called the q critical value (Lines 169-170).
Point 7: Please check lines 173-176 for clarity.
RE: Thank you for your comment; we made changes accordingly (Lines 186-187).
Point 8: Fig. 3 Please check that the figures are in the correct order. E.g., line 314 has WRSs as D
RE: Thank you for your comment. We have made the necessary changes (Lines 327-331).
Point 9: Line 392: IC is not defined, (move up from line 406)
RE: Thank you for your comment; we have made changes accordingly (Line 410).
Point 10: Please provide a reference for line 447.
RE: Thank you for your comment. We have made the necessary changes (Line 467).
Point 11: Abstract: line 22: did you run a correlation analysis? If not, remove ‘correlated’.
RE: Thank you for bringing this to our attention; we have made a change to address your comment (Line 23).
We hope that this manuscript is now suitable for publication in Brain Sciences.
Sincerely,
Prosper N'Gouemo
Associate Professor
prosper.ngouemo@howard.edu
Prof. Dr. Stephen D. Meriney
Editor-in-Chief,
Brain Sciences
RE: brainsci-2873060
Dr. Meriney
I have received reviews for the above-referenced manuscript. We are grateful to the referees for their valuable feedback and constructive comments. After carefully considering their suggestions, we have made significant revisions and clarifications as requested. We believe that the manuscript is now suitable for publication in Brain Sciences. All the changes made in the revisions and our responses to the referees are listed in detail below.
Revisions made by the authors
- We used the “Track changes” option for changes made in the manuscript.
- As suggested by Reviewer 2, we have excluded the following acronyms: WSP, WSF, IC, Nac, VTA, DR1R, and DR2R.
- We insert Supplementary Table 1 in the Results section (Line 281).
- We have included two new references (Lines 551, 637).
- We have made additional changes for enhanced readability and clarity: Lines 67-69, 86-87, 106, 108, 111-116, 186-187, 402-405.
Responses to reviews
Review 2
General comments
Point 1: The paper addresses the differences in voluntary alcohol consumption and withdrawal seizures in a mutant rat model, the genetically epilepsy-prone rat (GEPR-3s), and a control strain, the non-mutant sprague dawley rat. The association between genetic epilepsy and alcohol use and susceptibility is not well understood In addition to Strain, Sex and Age differences were also investigated. Overall, GEPR-3 seizure susceptible rats drank less alcohol than the control rats. When they did drink, GEPR-3 seizure outcomes worsened. Females drank more than males in both strains. The paper is well written and the data supports the conclusions. The hypotheses are clear and investigated in an appropriate way. Only minor edits need to be made before publication:
RE: Thank you for your positive feedback on our manuscript. We believe that your suggested modifications will significantly improve the revised version.
Point 2: As a general comment, there are many acronyms used throughout the paper, and it can be a bit confusing for the reader. I would suggest limiting the use of acronyms to the most frequently used terms only.
RE: "Thank you for your feedback; we have considered it and made the necessary changes.".
Point 3: Subjects: Please provide the total number of animals used in the study (is it 96?). Since you bred the animals for seizure susceptibility and also suggest generalized mutation effects on alcohol use I think it is important to include details about the parent generation and early offspring environment. What generation are these offspring? How are the parents paired? Do both male and female parents have seizures that are characterized? How are offspring caged at weaning/ when did you isolate them?
RE: Thank you for your comments. The total number of animals is 96, as mentioned in Line 73. The offspring GEPR-3s used in this study belong to the 45th generation of our colony (Lines 78-79). During the breeding process, we paired one male and one female in the same cage (Line 80). To ensure the presence of the genetic seizure trait in our colony, we only mate males and females that had generalized tonic-clonic seizures (Lines 80-81). After weaning at postnatal day 21, GEPR-3s were housed in pairs of the same sex per cage, as stated in lines 81-82. Adult animals are separated individually for five days (acclimation) before starting the experiments, as mentioned in Line xx.
Point 4: Methods: what is the purpose of the one week of water in between the drinking dilutions? It doesn’t seem very translational. Would these wash out periods affect subsequent withdrawal seizures?
RE: Thank you for your feedback. To minimize interference when measuring animals' ethanol intake, we implemented a 7-day washout period for different alcohol concentrations. We do not believe that these washout periods would impact subsequent alcohol withdrawal seizures.
Point 5: Please check line 145-146 for grammar affecting clarity.
RE: Thank you for your comments; we have made the necessary changes (Lines 155-157).
Point 6: Please provide a rationale or reference for preference of using a q-value.
RE: The Tukey test compares the difference between each pair of means of the groups with an appropriate adjustment for the multiple testing, called the q critical value (Lines 169-170).
Point 7: Please check lines 173-176 for clarity.
RE: Thank you for your comment; we made changes accordingly (Lines 186-187).
Point 8: Fig. 3 Please check that the figures are in the correct order. E.g., line 314 has WRSs as D
RE: Thank you for your comment. We have made the necessary changes (Lines 327-331).
Point 9: Line 392: IC is not defined, (move up from line 406)
RE: Thank you for your comment; we have made changes accordingly (Line 410).
Point 10: Please provide a reference for line 447.
RE: Thank you for your comment. We have made the necessary changes (Line 467).
Point 11: Abstract: line 22: did you run a correlation analysis? If not, remove ‘correlated’.
RE: Thank you for bringing this to our attention; we have made a change to address your comment (Line 23).
Round 2
Reviewer 1 Report
Comments and Suggestions for Authors
Thank you for answering my questions.
The manuscript underwent minor corrections. I have no more questions.